# Perspective and Evolution of Gesture Recognition for Sign Language: A Review

**DOI:** 10.3390/s20123571

**Published:** 2020-06-24

**Authors:** Jesús Galván-Ruiz, Carlos M. Travieso-González, Acaymo Tejera-Fettmilch, Alejandro Pinan-Roescher, Luis Esteban-Hernández, Luis Domínguez-Quintana

**Affiliations:** 1IDeTIC, Universidad de Las Palmas de Gran Canaria, 35017 Las Palmas de G.C., Spain; jesus.galvan101@alu.ulpgc.es (J.G.-R.); acaymo.tejera101@alu.ulpgc.es (A.T.-F.); alejandro.pinan101@alu.ulpgc.es (A.P.-R.); 2Signals and Communications Department, Universidad de Las Palmas de G.C. (ULPGC), 35017 Las Palmas de G.C., Spain; luis.dominguezquintana@ulpgc.es; 3Aries Research Center, Universidad Antonio Nebrija, 28015 Madrid, Spain; lesteban@nebrija.es

**Keywords:** gesture recognition, algorithms, Leap Motion, pattern recognition, EMG, RFID, gloves, Wi-Fi

## Abstract

This review analyses the different gesture recognition systems through a timeline, showing the different types of technology, and specifying which are the most important features and their achieved recognition rates. At the end of the review, Leap Motion sensor possibilities are described in detail, in order to consider its application on the field of sign language. This device has many positive characteristics that make it a good option for sign language. One of the most important conclusions is the ability of the Leap Motion sensor to provide 3D information from the hands for due identification.

## 1. Introduction

Gestures are a component of non-vocal communication, in which body language sends specific messages instead of or in addition to speech. They are made through the movement of different body parts, i.e., typically hands, arms and face.

Hand gestures are the most expressive and most frequently used, as they seem to be more natural and intuitive gestures for people in general. Their impact on the message of speech is extremely high. For example, within the context of emotional messages—according to researcher Albert Mehrabian—both signals and gestures can carry up to a 55% of the message impact [1].

Humans have been using this type of nonverbal communication for over a million years, and even some of these basic signals have become universal gestures: moving one’s head to affirm or deny something, frowning to express annoyance, shrugging one’s shoulders to indicate that something is not being duly understood, or even those gestures inherited from the animal kingdom, such as showing the teeth to express aggressiveness. Nonverbal communication has developed in such a way (see Figure 1), that some languages have been created from a series of gestures. Some examples are the sign languages used among deaf people or the universal signs used for aviation, diving or first aid.

Nowadays, technologies form a natural part of daily human life, thanks to a rising number of applications. The growing capacity in terms of computing power and speed for processors and sensors (Table A1), enables the existence of increasingly complex applications (Table A2) (Figure A1). Hence, the need for a more efficient human-machine interaction (HMI) is also proportionally increasing.

Within this context, on one side, the use of touch screens and voice interfaces is currently becoming consolidated, whereas on the other side, the keyboard-mouse exchange in favour of more natural and intuitive interfaces is still far from being a reality. This search for more direct interfaces has become an interesting option in those situations where the use of hands can represent some kind of inconvenience or difficulty when it comes to manipulate an input device directly. For example, when a driver wants to check the navigation system of his/her vehicle without taking his/her hands off the steering wheel, or when a person in the middle of a meeting wants to act discreetly with a computing device. Moreover, direct physical manipulation might spread microbes. Hence contact-free input mechanisms might become crucial in those spaces that cannot be contaminated in any way whatsoever, for instance, surgery rooms.

In this sense, gesture recognition may be considered useful in this path of increasing the interaction between machines and humans even more (Table A2). Since the rise of digital video capture technologies, there have been some attempts to recognize dynamic gestures for different purposes. Moreover, the update of new technologies such as depth sensors or high-resolution cameras enables the development of several ways to detect different movements and act in real time.

These types of data captures can be applied in multiple areas, such as patient monitoring, virtual and augmented reality navigation and manipulation, home automation, robotics, vehicle interfaces, PC interfaces, and lexicon translation of sign languages, among others (Table A2).

By definition, all these interactions consist of making gestures understandable for computers. The position of the human body, its configuration (angles and rotations) and its movements (speed or acceleration) must therefore be adequately detected. To perform this task, there are two main different approaches: the non-optical approach consists of using sensor devices such as gloves or bracelets, whereas the optical approach relies on the use of camera vision techniques. Both methodologies present a range of variations in terms of accuracy, calibration complexity, resolution, latency, range of motion, user comfort and costs.

Furthermore, it is necessary to make a clear distinction between gestures and poses. To start with, a pose corresponds to a static gesture, for example, a hand with no movement at all. In contrast, the gesture is a dynamic act formed by a sequence of connected poses during a short period of time, for example waving goodbye with a hand. Hence, gesture recognition might be challenged in two levels: a lower one demanding an adequate pose detection and a higher level regarding adequate gesture recognition.

Apart from this differentiation, interpreting both poses and gestures is not a simple task for various reasons. Firstly, there are usually several concepts associated with one same gesture and vice versa, and secondly, gesture meanings can vary between different languages, cultures and contexts. For example, sign languages used by deaf people vary between countries (sometimes even within the same country), these languages mix gestures and poses, and logically they might also vary depending on the conversation context.

## 2. Evolution of Gesture Recognition Devices

As mentioned above, in terms of human-machine interaction, gestural recognition is considered as the most intuitive and natural, so its development is constantly evolving, depending on the improvement of sensors used to capture gestures (Figure A1). In this sense, gesture recognition has essentially evolved from an intuitive recognition, to a more formal recognition based on the improvements from experiments on sensors used for this purpose. Initially, during the 1960s, researchers started using tablets and special pencils that captured writing (sketchpad [2]), touch-sensitive interfaces or pointing devices (see Figure 2). Later, in 1969, engineer Myron Krueger started working with virtual reality prototypes, and claimed that a future was possible without screens, in which people would interact directly with their environment [3]. This approach would later be referred to as natural interaction, and these interaction interfaces as natural user interfaces (NUI) [4].

It would not be until the 1980s when movement capture by wearing gloves with sensors for flexion and position would start. In the 1990s, works on identifying gestures in images and video with computer vision methods increased, something that has been improving until today where real-time and people tracking systems are available [5,6]. Apart from computer vision, there are also electromagnetic systems, which locate an object position by measuring the electromagnetic fields generated by a transmitter, for example, radiofrequency.

It would not be until the 1980s when movement capture by wearing gloves with flexion and position sensors started. In the 1990s, work on identifying gestures in images and video with computer vision methods was increased, and has been improving until today, where real-time and people-tracking systems are already available [5,6]. Apart from computer vision, there are also electromagnetic systems, which can locate an object position by measuring the electromagnetic fields generated by a transmitter, for example, radiofrequency.

### 2.1. Data Gloves

These devices are worn on the hands in order to measure their position and their movements. They usually have either tactile or other types of sensors incorporated, capable of producing an exact mapping of the movements performed by all different phalange and wrist joints. This has the great advantage that no data processing step for obtaining descriptors is needed, as would happen in the case of an image obtained with a camera, for instance. Its disadvantage is that they are expensive and uncomfortable for the user.

They can be divided into two distinct categories: actives and passives [7] (Figure 3). On one hand, active systems include all such gloves that include some type of sensor or accelerometer. Formerly, these types of data gloves are connected to the computer by cables, a feature that has already disappeared thanks to wireless technologies. On the other hand, passive or non-invasive gloves refer to non-electronic devices, which include certain colour markers in order to facilitate image identification.

The first type to appear was the Sayre Glove in 1977 and based its operation on flexible tubes (instead of optic fibre) to measure finger flexion by emitting a beam of light at one end, while a sensor (photoelectric diode) detected the intensity of that emitted beam according to the flexion and position of the fingers. 

In 1980, MIT Media Lab started using gloves similar to Sayre’s, only they were more interested in developing a movement capture device, rather than a control instrument. Unfortunately, the technology was not yet developed enough to obtain a truly effective input device, so the use of this glove shortly came to an end [8]. 

In 1983, the first glove that actually recognized hand positions with the intention of creating alphanumeric characters was patented, and the first virtual reality glove was marketed only four years later [9]. Physically, this DataGlove consisted of a lightweight glove with optical fibres attached to the back of the fingers. In this model, the flexion of the fingers bends the fibres, thus attenuating their transmitting light. The signal intensity for each fibre is duly sent to a processor, which then determines the angles of the joints based on the pre-calibrations for each user. 

Later, in 1989 and 1995, the Power Glove and Super Glove appeared, and they both used materials of variable electric resistance in order to measure the finger flexions. These were cheaper but at the same time less accurate [10]. Once the use of magnetic sensors and accelerometers were added, numerous examples of these devices started to appear, both for commercial and scholar purposes. Currently, the CyberGlove [11] (Figure 4) is the most acknowledged model, and has been particularly developed for the graphic animation and film industries. In addition, the MIT AcceleGlove [12] is being widely used for virtual reality frameworks.

The evolution in experiments on data gloves has enabled their use in different fields, ranging from classic applications such as sign understanding, or entertainment and virtual reality, to other fields such as medicine [13], robotics or industrial production [14].

### 2.2. Electromyography (EMG) Electrodes

Exactly the same way as active gloves, bracelets also try to avoid environmental interferences that voice interaction or computer vision systems might suffer. However, its operation is not based on accelerometers or optical fibres; it relies on electromyography (EMG) sensors. This means that it seeks to measure the electrical potentials produced by muscle cell activity. 

Among these recognition technologies, active research is being conducted on studies based on bio-signal recognition technology: electrooculography signal (EOG: electrical activity of the eyeball movements or “gaze position” recorded around the eyes), electroencephalography signal (EEG: electrical brain activity recorded from the scalp), electrocardiogram signal (ECG: electrical activity of the heart), EMG and others.

Electromyography is based on the study of the neuromuscular system, so it is responsible for detecting, analysing and processing the electrical signals produced by muscles and nerves, by using electrodes.

Muscles are composed of a set of specialized cells capable of contracting and relaxing, so they are responsible for functions such as producing movement, transporting substances through the body or providing adequate stabilization and temperature [15]. In addition, three types of muscle tissues can be differentiated according to their structure, contractile properties and control mechanisms: (a) skeletal muscles, (b) smooth muscles and (c) heart muscles. Skeletal muscles—according to its own name—are attached to the skeleton and facilitate body movement, while smooth muscles are inside the intestines and are responsible for the transport of substances, and finally, heart muscles are in charge of producing heartbeat. The EMG signal is obtained from skeletal muscles [16].

It must be noted that bioelectric potentials generated by the organism are actually ionic potentials produced by flows of ionic currents. The efficient measurement of these ionic potentials requires that they must be therefore converted into electric potentials, before they can be measured with conventional methods. Devices responsible for this conversion are the electrodes obtaining intramuscular or surface voltages [17]. EMG electrodes are capable of recording potentials from all the muscles within their reach. This means that the potentials of nearby large muscles can interfere during attempts to measure the EMG of small muscles, even when the electrodes are placed directly on such small muscles. Therefore, in these cases, the obtained signals are usually the result of a sum of all these individual potentials created by the fibres that form up the muscle or muscles that are being measured. When this might cause a problem, needle electrodes directly inserted into the muscle are required, as these can access muscle fibres individually. Consequently, intramuscular or needle electromyography is used in order to study the physiology or pathologies of motor units, whereas the surface type (SEMG) is more suitable for studies on types of muscular behaviour, patterns of temporal activity or muscular fatigue [18].

Factors such as the distance between electrodes, the area of muscular activity, the skin properties, the signal processing and the contact that occurs between the skin and the electrode are essential for analysing EMG signals, since amplitude and its properties depend directly upon all these.

Typically, signals obtained by the electrodes must be amplified and processed (Figure 5). At the same time, signal amplitude and bandwidth might vary depending on the size, type of electrodes, and the spacing between them [19].

Amplitude of an EMG signal ranges from 10 µV to 5 mV and its bandwidth from 10 Hz to 10 KHz, depending on the electrode configuration. Commercially, it uses silver–silver chloride electrodes (Ag–Ag Cl), due to its stability and noise reduction.

As for the number of electrodes needed, a minimum of three is required. Two of them comprise a differential input pair and the third one corresponds to the grounding. Sometimes, active or activated electrodes are added in order to create a feedback control circuit between the sensor and the body [20], thereby eliminating the need of using conductive gel between electrodes and the skin (its use decreases the electrical impedance).

Traditionally, EMG technology has been employed for medical diagnosis [21], prosthetic control [22] and medical rehabilitation technologies [23]. Within the human–computer interaction (HCI) context, a series of studies have emerged focusing on the use of EMG inputs as an alternative channel for users with physical disabilities, given that the activity in one or more muscles can be interpreted in a Graphical User Interface (GUI) both on a continuous basis (as a substitute for the mouse) or on a static basis by gesture recognition [24,25,26,27]. Other examples of EMG applications for HCI include robotic control [28,29,30], non-voice speech recognition [31], emotional state recognition [32,33], musical expression interfaces [34,35,36] and generic gestures recognition, using these for either controlling a music player [37], as a GUI pointer [38] or as a numeric keypad [39,40].

Regarding existing products in the market, due to recent developments in wireless communications and integrated computing, we can find widespread portable devices used to obtain EMG data using bracelets [41,42]. These include multiple EMG sensors radially positioned around the circumference of a flexible band, enabling a relatively comfortable fitting. An outstanding example might be the Myo bracelet produced by Thalmic Labs company, which has been used in many applications and experiments [43,44,45,46,47].

### 2.3. Ultrasound

Two techniques can be distinguished for gesture capturing. One of them uses ultrasound images (sonomyography), and the other that takes advantage of the Doppler Effect, profiting from the ultrasound radiation in any room. This technique, opting for the use of ultrasound images, enables the possibility of the observation of the muscles inside the body on a real-time basis.

This is achieved thanks to the fact that our tissues bear different acoustic impedances, so when a sound wave passes from one to another with a different acoustic impedance, different amounts of energy are reflected, thus forming an ultrasound image. It will unlikely provide greater accuracy than other systems measuring directly the moving parts of the body, such as cameras or data gloves, but instead, as in EMG techniques, potential benefits concerning information losses due to occlusion can be obtained, since in this case, any part of the body does not remain hidden by others. Additionally, and in contrast with EMG, sonomyography does not experiment with the difficulty of differentiating between individual muscles (for non-invasive techniques), and those lying deeper in the forearm, which limit the soundness of EMG techniques for capturing a variety of wide hand movements [48].

Various studies have emerged in this regard: from Hodges in 2003, measuring muscle contraction of several muscles [49], to Zheng and Chen in 2006, 2008 and 2010 [50,51,52], or Mujibiya, which in 2013 put forward a band of ultrasonic transducers around the forearm and fingers [53], Hettiarachchi which differentiated six gestures in 2015 [54] and McIntosh in 2017, which produced a system that detects and monitors a series of finger gestures using a probe fitted on the forearm [55].

Another technique using ultrasonic frequency signals for gesture capturing is based on the management of the Doppler Effect. It essentially consists of a transmitter that emits ultrasonic continuous tones, which bounce off of objects (arms and hands in gesture recognition) within the detection field. Reflected signals returning to the sensor are then captured for their due analysis. In a static environment, the returning signal bears the same frequency, but has a different phase and amplitude. However, if an object moves, its echoes shift their frequency (Doppler shift), thus creating components in other frequencies which are proportional to their speed in relation to the sensor.

The Doppler shift caused by reflection from a moving target is approximately *fd* = 2*vf*/(*cv*), where *fd* is the frequency offset, *v* stands for the target’s velocity (in the direction of the sensor), *f* shows the emitted frequency, and *c* represents the speed of sound [56].

In the case of movements performed by several objects with different speeds, the return signal will contain multiple frequencies, one for each moving object.

This is the same principle of radar operation used by the police for speed controls. However, ultrasound is used here instead of RF, and the ultrasonic sensor observes the full velocity profile in the field of view. In comparison, police radars can only detect the speed of a single object [57].

Although ultrasonic ranges have been considerably exploited in the market [58], research on this technique applied towards ultrasound gesture recognition has not been widely reported.

The first accounts in this area date back to 2009, when hand recognition was developed and was considered an important issue according to [59], in particular, when it is based on the signal power being reflected into three receivers (3D). A year later, a simple ultrasonic gesture recognition system for music control [60] was reported. In 2011, Kreczmer reported gesture recognition using the conventional ultrasonic range finder in a mobile robot [61]. Shortly thereafter, Microsoft Research and the University of Washington developed a nearby gesture ultrasonic recognition system for laptops [62]. That same year, a Berkeley group designed an ultrasonic gesture recognition system that uses a matrix with a large number of capacitive ultrasonic transducers on a single chip [63,64]. This research line continued developing, looking for a more compressed hardware with less spaced sensors, like the works in 2014 by Booji [65], or in 2017 by [66] or Yu Sang in 2018 [67].

### 2.4. WiFi

With the aim of re-using existing Wi-Fi infrastructures, which would save the corresponding employment of a hardware and with the possibility of an easy large-scale deployment of systems, the search for gesture recognition through Wi-Fi networks (Figure 6) began. An additional advantage of this technique, which shall be furtherly detailed in the cameras section, consists of its capacity for providing gesture recognition even under non line of sight (NLOS) scenarios (without direct vision). 

Moreover, different indicators are used on these type of systems to obtain all the information necessary for gestural recognition. For instance, some research studies are based on the received signal strength indicator (RSSI) [68], on the signal flight time indicator (ToF) [69] or on Doppler shifts [70]. However, these systems require either specialized devices [69,71], or need modifying by already-existing commercial devices [70], and some of them are excessively susceptible to interference [68]. Consequently, research has currently focused more on the observation of the channel status information (CSI). This information provides more accurate indications as it comes from a single signal flow, due to the orthogonal frequency division (OFDM). As a result, CSI has proven to be a more robust indicator even with indoor interferences [72]. 

Based on the RRSI use, Abdelnasser et al., [73] created a gesture recognition system using Wi-Fi (WiGest). This system can identify several hand gestures and map them for monitoring various actions, with accuracy rates ranging between 87.5% and 96%. For this task, it uses a single simple access point and three aerial ones. Other researchers like Pu et al. [70] introduced the WiSee system, which can recognize nine body gestures that interact with Wi-Fi-connected home devices, by taking advantage of Doppler changes in wireless signals. On the same line, Adib et al. introduced WiVi [71], Witrack [69] and WiTrack0 [74]. These systems can track human movement through the walls and classify simple hand gestures. Concerning CSI, Nandakumar et al. [75] presented a four-gesture recognition system with a 91% success in LOS scenarios and with 89% in NLOS. In contrast, El WiG of He et al. [76] classify the same number of gestures but with a 92% and an 88% success respectively. Some other research works include more complex gestures, such as the WiFinger by Li et al. [77] which identifies considerably subtle finger gestures with a success rate of 93%.

### 2.5. Radio Frequency Identification (RFID)

RFID systems generally consist of ultra-high frequency (UHF) commercial readers, which are capable of detecting labels within a range of even a few dozen meters, depending on the transmitted power. Identification tags can either be carried by a person or not, depending on the RFID application, and they can work with or without a battery (active and passive). Normally, RFID tags have been used for supply chain management and automatic object identification [78,79]. They also have been used to increase the digital information environment [80], to monitor indoor human activities [81], or to detect human interactions with RFID-tagged objects [82]. Just like in the case of Wi-Fi signals, relevant information such as phase changes, RSSI and Doppler shifts [83] can be collected from UHF signals, thus providing potential possibilities for gesture recognition. 

An advantage of using passive RFID technology is its inexpensive costs but—logically, since they do not count with their own energy supply—detection ranges are reduced to centimetres. Due to this inherent setback, related gesture detection research is very focused towards detecting interactions between humans and objects or tactile interfaces [84,85,86,87]. Active tags, due to their long range, are normally used in traceability, location and identification applications.

### 2.6. RGB and RGB-d Cameras

Nowadays the most widely used devices for gesture captures are definitely RGB and RGB-d cameras [7]. Current digital cameras capture the light in three main channels of red, green and blue light (hence the acronym red (R), green (G) and blue (B)). The resulting image is organized in an array of pixels, each one bearing its three RGB values. Therefore, any attempt to identify an object within the image will result in a computer-vision algorithm, in which, essentially, the order and intensity of colour of the different pixels will provide a meaning.

Depth cameras, also known as RGB-d, are a variation of conventional cameras. These have a fourth channel for showing the depth information, i.e., the distance between the focus and the captured object. In order to obtain this depth information, and therefore produce 3D images, three clearly differentiated techniques are used.

#### 2.6.1. Stereoscopic Vision

This is achieved by placing two cameras at par values to obtain two different views of the same scene. This process is similar to human binocular vision procedures. In order, however, to obtain such depth, a series of calculations must be performed on the obtained pair of images. Since two pixels actually correspond to one same point of the scene, its depth can be measured by knowing the system calibration conditions of such parameters, like the focal length or the separation between both camera centres [88].

#### 2.6.2. Structured Light

This consists of an active illumination of the scene by projecting a pattern that varies spatially in the *x* and *y* coordinates. Therefore, it is considered a modification of a stereoscopic vision, which replaces the use of a second camera in favour of a light source [89].

#### 2.6.3. Time of Flight (TOF)

These cameras can estimate the 3D structure directly, without the help of any artificial vision algorithms. Its operation is based on the emission of a modulated light source, usually within the infrared (IR) spectrum, which illuminates the scene. Distance is obtained by measuring the flight time of the signal that travels from the light source, collides at the scene and returns to the sensor element [90]. 

The problem of designing a system capable of recognizing gestures for controlling applications usually arise in the following stages (Figure 7): data collection and processing (camera capturing, processing and segmentation), modelling of the captured object (selection and management of object features), definition of gestures (static and dynamic), their detection (posture separations and motion detection) and transition control (state machines).

Each of the previous sections produces a range of research studies, depending on the processing modality, chosen segmentation and classification, and the possible use of multiple cameras to deal with the problem of objects occlusion (hands and arms in the case of gesture capture) [91,92].

Some outstanding examples are Cohen and Li [93], which classify body postures with the support vector machine (SVM) technique of a 3D visual helmet, built from a set of data entry. This system returns classified positions of the human body as miniature images. Moreover, ChengMo et al. [94] have pointed out a human behaviour analysis by using a system that recognizes human postures such as walking, bending down or sitting. Moreover, Corradini et al. [95] could analyse and recognize positions by using hybrid neural networks. Furthermore, Raptis et al. [96] for instance, presented a classification system which recognizes dance gestures based on the movement of six joints, collected on a real time basis by a Kinect camera with a 96.9% accuracy.

Although these 3D cameras have been used for gesture recognition systems, traditionally they have been expensive [97]. Since the appearance of the Kinect Microsoft device, costs have been significantly reduced, thus becoming a very attractive option for many researchers [98]. The system returns the classified positions of the human body as miniature images [99].

Recently, Microsoft released a new version of the Kinect device, called Azure Kinect DK. Although the previous version was mainly focused on games; this new device is focused on professional use, covering markets such as logistics, robotics, and healthcare, among others. Azure Kinect DK is a developer kit and a PC peripheral that uses advanced artificial intelligence (AI), sensors for sophisticated computer vision and speech models. It combines a depth sensor and spatial microphone array with a 4K RGB video camera and an inertial measurement unit. The whole device is applied with multiple modes, options, SDKs, and Azure cloud services support. 

Microsoft has also provided a new skeleton tracking kit that detects the 3D coordinates of human body joints. The body-tracking features provides body segmentation, an anatomically skeleton for each partial or full body in the field of view, a unique identity for each body, and can track bodies over time. 

It is a device with a high cost that requires fairly considerably minimal requirements. Additionally, it is a recent device and no scientific publications related to sign language have been found, but rather some proprietary solutions from private companies.

### 2.7. Leap Motion

The Leap Motion [100] (Figure 8) is a compact and affordable recognition device, which is capable of 3D tracking forearms, hands and fingers in real time. Its configuration is oriented to interactive software applications.

It contains two infrared cameras (Figure 9), with an angle of 120° and three infrared LEDs, so it would be classified like a ToF camera, according to the previous section on camera types. These sensors work at a wavelength of 850 nanometres, i.e., they work within a non-visible spectrum for the human eye, and use sample speeds of up to 200 fps (frames per second), adapting their lighting to detected light, clarifying the flood detection zone and ensuring a constant image resolution.

As can be observed, small plastic barriers are placed between these LEDs and the cameras, in order to avoid interference and saturation, and to achieve uniform illumination throughout all covered areas. 

In addition, sensors are CMOS type, which means that no external electronics are required, as the digitization of each pixel occurs within each cell. All this results in a faster capture speed, using less hardware space [101]. 

Moreover, the microcontroller (Macronix MX25L3206E [102]) performs tasks such as lighting regulation, obtaining the information collected by the sensors and sending it to the USB controller.

Additionally, the USB controller controls the data transfer to the computer to which the device is connected. It is high speed and can support USB 3.0. Moreover, sent data are received by the computer controller through two serial ports, UART_Rx and UART_Tx.

The interaction area (Figure 10a), forms a semi-hemisphere with a viewing angle of 150° and with an effective range that extends from 25 mm to 600 mm. All these parameters delimit the interaction scope (interaction box). The interaction height range (see Figure 10b) can be configured between 70 mm and 250 mm.

This area can be influenced by several factors such as the camera viewing angle or the maximum intensity that the USB can deliver. It should be noted that the viewing angle (1) is also affected by the focal length and sensor size, in the following way: “*d*” is the sensor diagonal and “*f*” stands for focal length.
(1)$$\propto =2*\;\tan^{-1}\left(\frac{d}{2f}\right)$$

It has a USB connection and is compatible with Windows, Linux and Mac operating systems. Furthermore, the company provides APIs for developers to program this device in Python, Java, C ++, C #, Objective-C and JavaScript, finally evolving to Leap C, a C style to access tracking data of the Leap Motion service. It also distributes plugins for the Unreal Engine and Unity graphics engines.

Here, the controller analyses each image looking for hands and fingers, and sends a set of frames to the computer via USB. Each frame object (Figure 11) contains Hand-type classes with its correspondent Arm subclasses (a bone-like object that provides the orientation, length, width and endpoints of an arm) as well as Finger subclasses, which contain data concerning identifying types, direction position, orientation, speed, length and width, angle between them and even percentage of coincidence with the internal model contained by this device. Besides all of these parameters, it can also recognize certain predefined dynamic gestures (Gesture) and tools (Tool) [103].

The internal model of a human hand is used to provide a predictive tracking even when parts of a hand are not visible (due to occlusion). The software uses the visible parts of the hand, its internal model, as well as past observations to calculate the most probable positions of non-visible parts at any given time. A *Hand.confidence* () rating indicates how observed data fit the internal model [104].

Besides the hand skeletal tracking model (Figure 12a), Leap Motion also provides a screen, where the recorded sequences of a scene happening in front of the device can be observed in real time (Figure 12b). As mentioned earlier, this occurs thanks to an infrared radiation (IR) of the scene, where its reflection is captured by biconvex cameras. Data obtained by the sensors are stored on a digitized image.

As images are obtained, a distortion produced by the lenses can be observed (see Figure 13) on these optical images, thus deforming them. This distortion produced by the Leap Motion is known as complex distortion and is a mixture between barrel distortion and cushion distortion [105].

This distortion can be calibrated (Figure 14) by superimposing a mesh map of calibration points on the captured image. As pairs of image data are being sent to the microcontroller, each value is accompanied by its distortion information. Finally, the correct brightness value can also be achieved by using a calibration map [106].

Once the corrected images have been obtained and the microcontroller has identified the hands and fingers in the interaction zone, their position in the coordinate system is determined through stereoscopic vision techniques [107].

There are different studies that have checked this device accuracy and its robustness [105,108,109,110]. It has been evidenced that it obtains better results than other contemporary devices of the Kinect or Myo types. Additionally, both static and dynamic measurements have also been performed to analyse systematically the controller sensory space and define the spatial dependence of its accuracy and reliability. During static experiments, when objects move away from the device and are located at the right or left ends, a significant increase in standard deviation is observed. Concerning dynamic experiments, a significant drop in sample accuracy was observed when these were taken over 250 mm above the controller.

Due to the Leap Motion design, mainly oriented to applications such as virtual reality (VR), gaming or more natural interfaces use (NUI), we can find many publications in this regard [111,112,113,114].

It is not hard to imagine that the features of this device are suitable for research outside of the VR and gaming fields. This fact is evidenced by the large number of publications concerning both static and dynamic gesture recognition using Leap Motion. For example, in 2014, Mohandes et al. [115] managed to capture 28 letters of the Arabic sign alphabet with a 99% success rate, by using multilayer neural networks (MLP) with the Nave Bayes classifier. The following year, Chen et al. [116] using an SVM algorithm, captured 36 gestures (numbers and alphabet) with a success rate of 80%. That same year, McCartney et al. [117] achieved an accuracy rate of 92.4% in a database of over 100 participants, using convolutional neural networks and hidden Markov models. A year later, Wei Lu et al. [118] managed to recognize a set of dynamic data with a success rate of 89.5% by using neural networks as a classifier (HCNF).

Logically, in the same way as other technologies explained above, the use of Leap Motion has extended to other areas such as robotics [119], medical rehabilitation [120], home automation [121], identification and authentication [122], music [123] or education [124].

Currently, the same company that produces Leap Motion has released a new device called UltraLeap. This device is more accurate and powerful than the Leap Motion. It started to be commercialized a few months ago with a very high price. There are no scholarly papers yet concerning this device, nor any scientific application for it.

### 2.8. Virtual Reality (VR)

Nowadays, the advances based on resolution, which is included on the latest VR-based devices, help to create interesting interactive spaces for different fields, which need to be added to finger tracking, sensations or custom environments. This benefits applications that are aimed at medical rehabilitation, education, professional training and virtual practices of the architectural type, virtual test benches for industrial design and engineering, teleoperation of robots [125], etc. Efforts have been focused on creating spatial awareness, as real and accurate as possible, to create a full user experience. Therefore, users want to achieve the head-mounted display (HMD), a mounted screen that leads to the highest possible degree of presence. In the case that the HMD achieves this objective, it is understood that the performance of the trace is improved. This is explained in [126], where it proves that due to the Oculus Rift’s support for Leap Motion, the user performance is significantly better in virtual reality settings. It requires combinations of sensor hardware and processing software, based on different detection technologies such as depth cameras, radio triangulation, ultrasound, IR reflectance of mobile screens, polarized light, or marker-based tracking with an IR camera. The tracked movements of the user are transformed into arbitrary coordinate systems and data formats; therefore, the VR environment serves as a kind of meta-layer to integrate and simulate different virtual tracking systems. [127].

The infinity of possibilities that VR offers has prompted companies like HTC, Oculus, Valve, HP, Microsoft and Sony to manufacture their own glasses. Striking examples are the alliance of HP, Valve and Microsoft to launch the HP Reverb G2 [128] to achieve 90 Hz resolution, which is four times higher than Oculus Rift [129] and HTC Vive [130]. Additionally, we highlight the Valve Index glasses [131] with their experimental 144 Hz mode, which increases the field of view 20° more than HTC.

## 3. Discussion

Since the Sketchpad in the 1960s to the present, there has been significant research and strong efforts regarding the production of devices that are capable of recognizing gestures. The most important evolution has been observed since the 1980s, when improvements in semiconductor technologies enabled the creation of sensors recognizing different hand movements by transferring them to control systems. A fairly widespread example has been the use of different types of gloves, (DataGlove, PowerGlove, SuperGlove, CyberGlove, AcceleGlove). As devices, it should be noticed that these examples started being invasive and not very versatile, although it is true that later, their quality and accuracy improved, over the years. This improvement has been achieved not only due to the type of sensors that have been used, but also due to software development regarding information management. 

Moreover, many video processing techniques have tried to recognize movement. Their main hindrance compared to other techniques is the fact that these always work in two dimensions and the recognition of gestures requires 3D systems.

In the second decade of the twenty-first century, the Kinect was released. This milestone represented an important revolution in the recognition of 3D gestures. The Kinect sensor belongs to the group of ToF cameras, and is able to detect the movements of a person and transfer—in real time—the contour of that person along with his/her movements. This device was not specially designed for the detection of hand movements; therefore, it is not precisely the most ideal device to be used with sign language. In addition, their size reduces their versatility considerably as a transportable device.

Since this mentioned decade, other types of components have appeared, such as the Leap Motion, included in the ToF component category, capable of recognizing the movements of the palm of the hand in 3D. This device was launched into the market with the aim of handling any type of environment simply by using hand movements. Research performed on Leap Motion tested the power of this device concerning the detection of different types of static gestures with percentages of up to 99% accuracy. Its dynamic use has also produced very high recognition percentages in different research studies. These properties of Leap Motion make it an ideal device for the recognition of sign language.

The rest of the technologies used for the recognition of gestures, for example, those related to the different types of sound waves (Wi-Fi, Ultrasound and RFID), are not able to detect the hand and finger movements with accuracy. In addition, there are no specific devices for these types of other technologies. Research in these fields mainly focuses on how the Doppler effect can be managed.

## 4. Conclusions

The evolution of different devices and their application for sign language purposes has been an active matter since the 1960s, and continues to be on a constant basis up until today. It is an open knowledge area where technological solutions are being increasingly approached, but it needs some more thorough consideration and development. Interest in this field is certainly significant, as well as the amount of studies regarding this research line.

There are many devices that are capable of recognizing gestures. Some of them have excessive complexity and size, and they are subordinate to using different active elements. An example may be the use of both active and passive gloves. Other devices have complex data post-management and some of them even have low recognition rates, also.

In contrast, the Leap Motion has many advantages over other devices. In the first place, this technology is very cheap, its usage and transport are easy too, it clearly recognizes your hands and you can work with it on multiple platforms. It can also be integrated into electronic devices and, most importantly, no active element is required. The manufacturer also provides all the necessary software to process—in a simple way—all the generated information.

Finally, their interaction area within their reach is more than sufficient, so that they can recognize, for example, sign language.

## Figures and Tables

**Figure 1 sensors-20-03571-f001:**
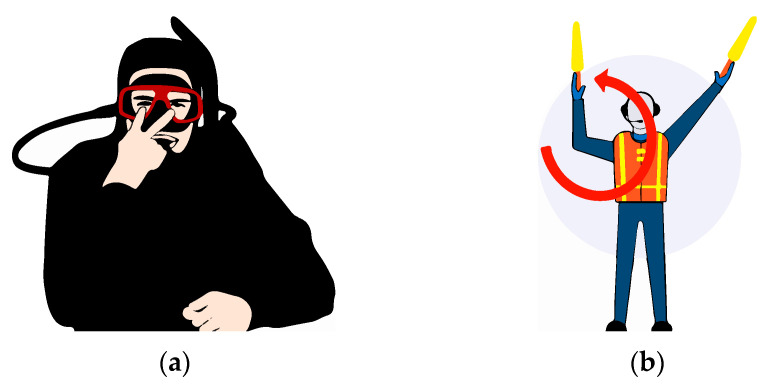
Sign languages for different uses. (**a**) Diving Sign. (**b**) Aviation Sign.

**Figure 2 sensors-20-03571-f002:**
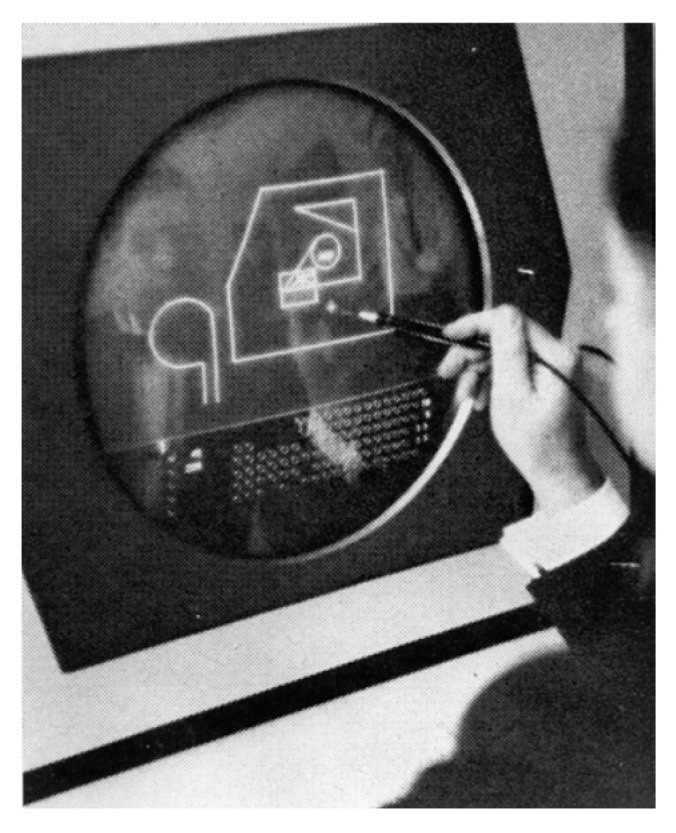
Written on sketchpad in 1963.

**Figure 3 sensors-20-03571-f003:**
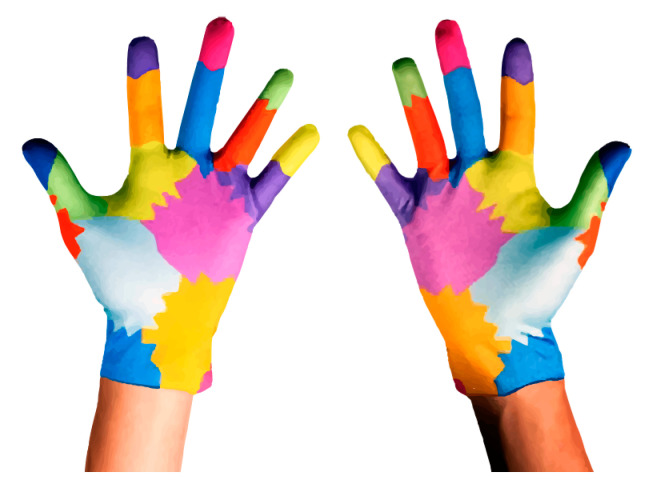
Passive gloves to help differentiate finger position.

**Figure 4 sensors-20-03571-f004:**
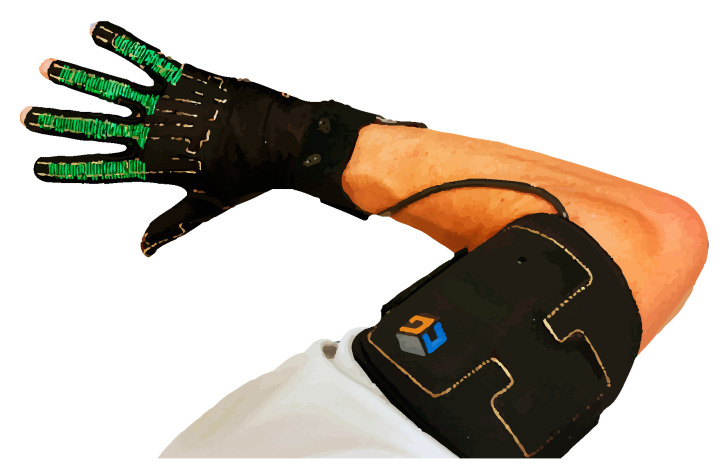
CyberGlove III mainly used to control robots.

**Figure 5 sensors-20-03571-f005:**
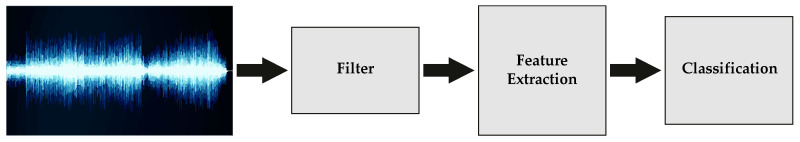
Different phases for the treatment of electromyography (EMG) signals.

**Figure 6 sensors-20-03571-f006:**
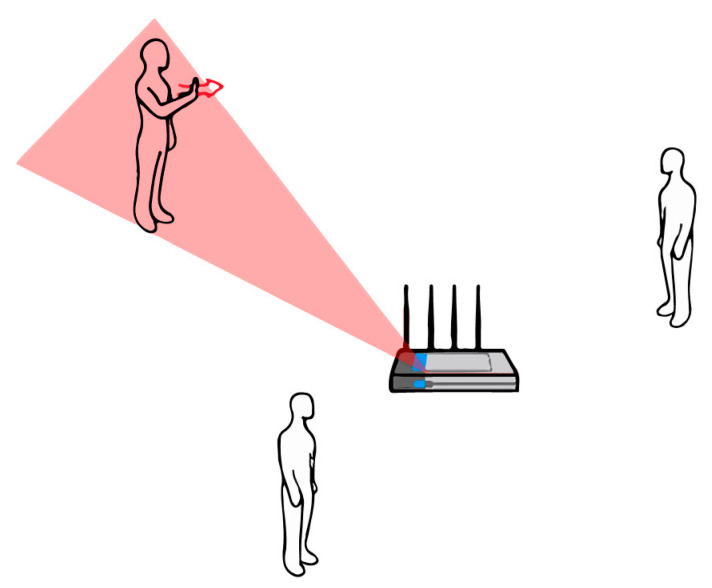
WFI recognition of different hand positions.

**Figure 7 sensors-20-03571-f007:**
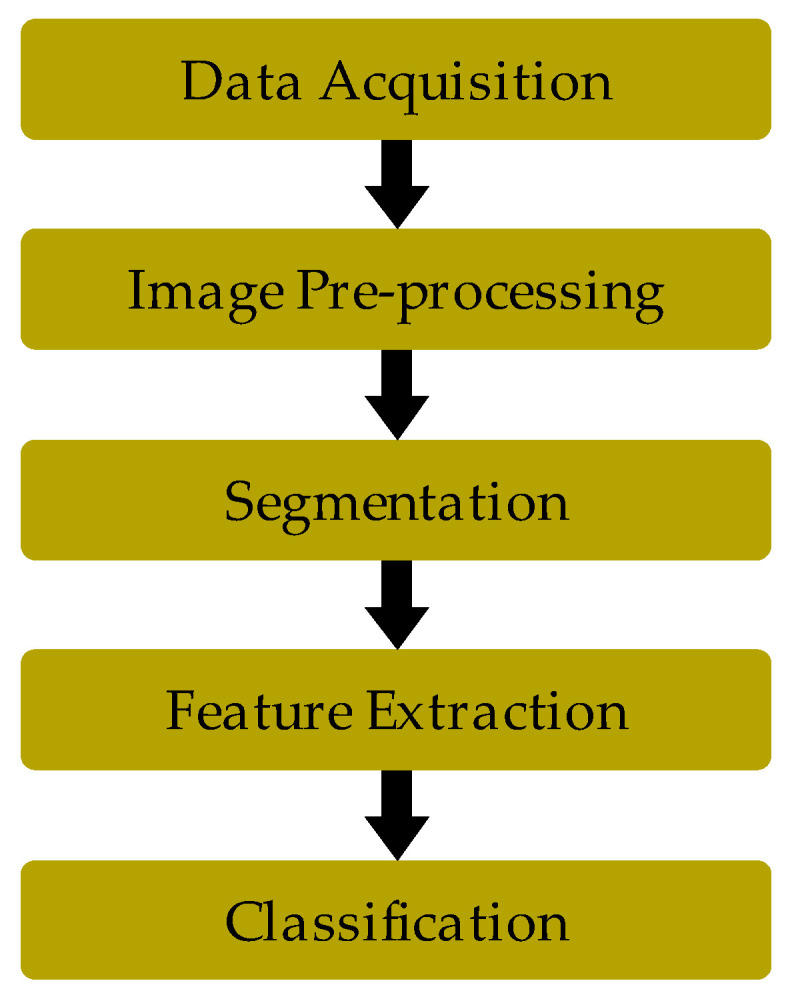
Stages of vision-based systems.

**Figure 8 sensors-20-03571-f008:**
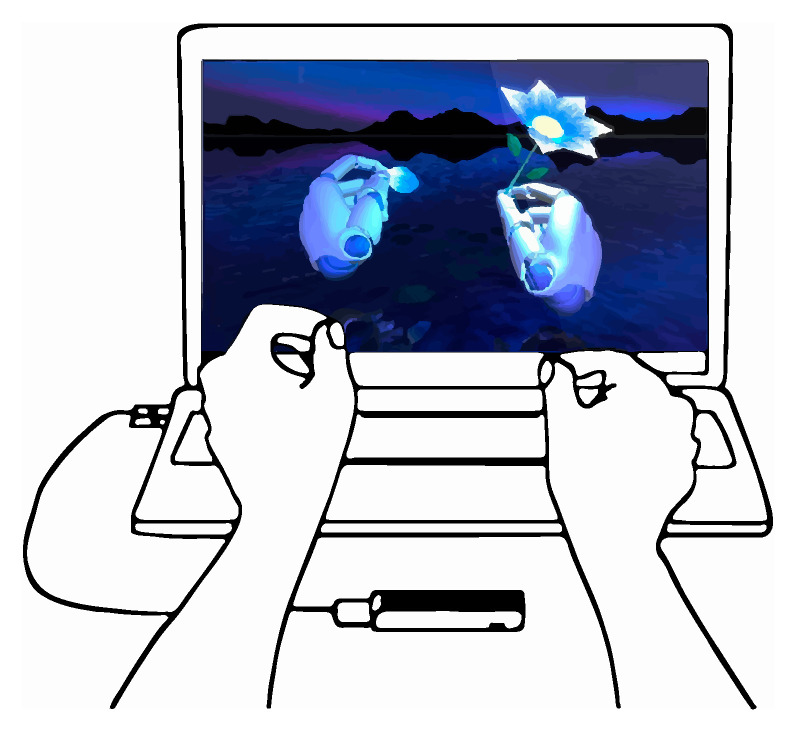
Leap Motion operating mode.

**Figure 9 sensors-20-03571-f009:**
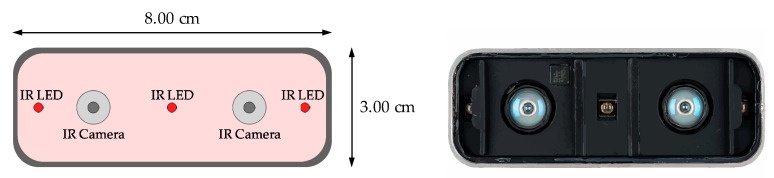
Different components of the Leap Motion hardware system.

**Figure 10 sensors-20-03571-f010:**
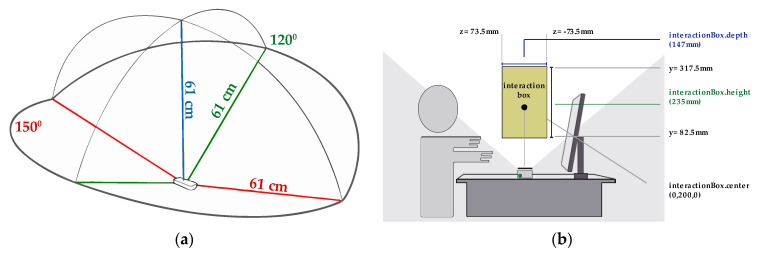
(**a**) Interaction area of Leap Motion. (**b**) Interaction box of Leap Motion.

**Figure 11 sensors-20-03571-f011:**
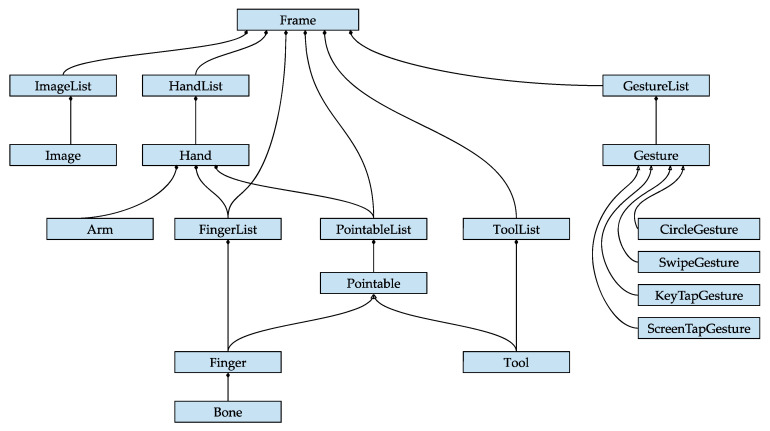
Frame Object of Leap Motion Source.

**Figure 12 sensors-20-03571-f012:**
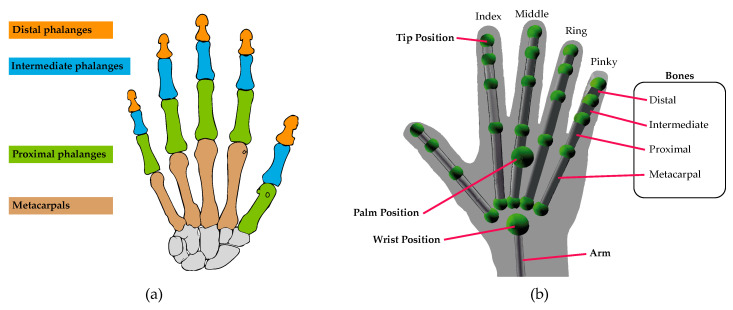
Tracking of the hands skeletal model (**a**) Object Bone. (**b**) Object Hand.

**Figure 13 sensors-20-03571-f013:**
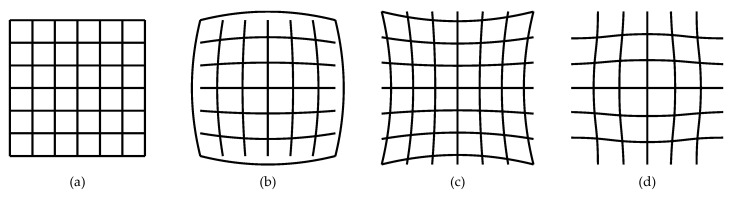
Distortions that lenses can produce (**a**) Without distortion. (**b**) Barrel distortion. (**c**) Cushion distortion. (**d**) Complex distortion.

**Figure 14 sensors-20-03571-f014:**
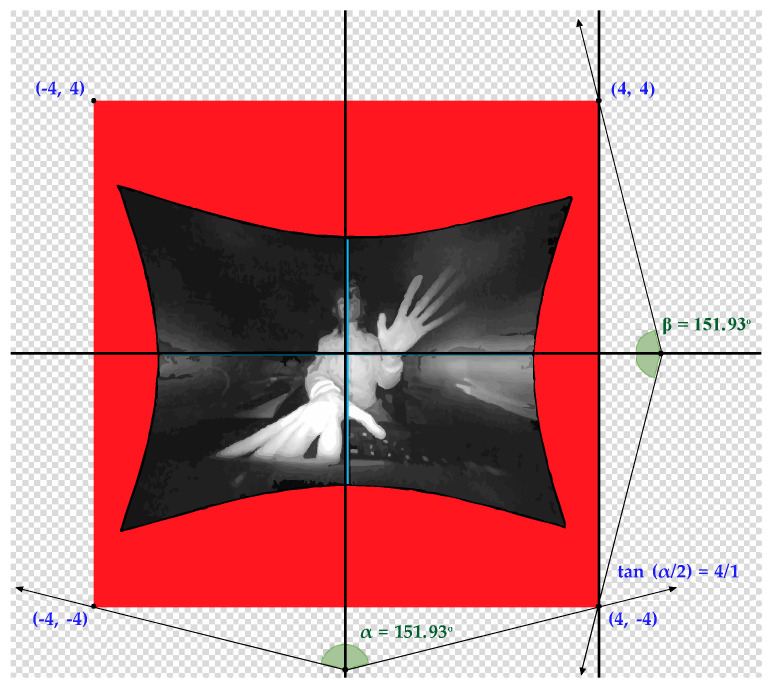
Image with the distortion corrected before reaching the microcontroller.

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
