# Peer review of "Perspective and Evolution of Gesture Recognition for Sign Language: A Review"

_sensors, 2020, doi:10.3390/s20123571_

Round 1

Reviewer 1 Report

No further comments

Author Response

Authors are thankful to Reviewer, who have given their valuable suggestions to improve the quality of the paper

Reviewer 2 Report

The paper overviews the field of gesture recognition — a field that is gaining on popularity, and arguably will continue to, especially with the recent advancements in virtual reality and the popularity of natural user interfaces. I like especially Appendix, where Tables 1 and 2 survey the field very well. However, the major drawbacks that authors should address before publishing are below:

- Valve Index and other new VR-based setups that offer full finger-tracking need to be added before publishing this paper.
- The paper offers great review. However, Section 2.7 could be shortened a bit more and perhaps add a bit more on Kinect (the authors do mention and refence Kinect though, but briefly). Furthermore, the new version of Kinect is missing.
- Figures' captions are rather short, I think it would work better to have longer, more descriptive captions.
- Figure 11 is not necessary for the paper.
- There are two instances of "Neuronal network".
- Table 2 of the Appendix (the core of the paper, but needs a bit of polishing):
- Separators (e.g. using double line) could be used between Device Types, e.g. when all the lines of Ultrasound end and Wifi begins, etc.
- The first line of Table 2 does not fit well according to the Table caption "All devices used to recognize both static and dynamic gestures". The change of caption into "works" (or something similar) instead of "devices".

Round 2

Reviewer 2 Report

The authors addressed the comments and the paper can be published. However, before publishing, I would kindly ask authors to do the following:

  • There are still two occurrences of "Neuronal" networks, please, fix this to "Neural".
  • You removed Figure 11, which is great, because it was unnecessary, however, fix the Figure numbering which go from 10 to 12 now.
  • In Table 1, for Kinect, please add disclaimer where it says "(Currently on sale)", that this is for version V2.

Author Response

This manuscript is a resubmission of an earlier submission. The following is a list of the peer review reports and author responses from that submission.

Round 1

Reviewer 1 Report

The paper overviews the field of gesture recognition — a field that is gaining on popularity, and arguably will continue to, especially with the recent advancements in virtual reality and the popularity of natural user interfaces. The fact that the paper overviews such an important topic makes it important and very useful for the future.

- The paper offers great review. However, I think too much emphasis is put on a specific device — Leap Motion from a defunct company (the device is now called Ultraleap, I think). The authors should shorten Section 2.7, or if this is not possibile, add detailed descriptions like this for other devices on the market os well.
- The main contribution of the paper lies in the Appendix, with the Table 2. This should be more emphasized in the begining of the paper.
- Language needs polishing, especially the abstract, which begins with "This review wants to analyze ...", change to "This review analyzes ..." etc. However, language in the text gets much better. There are still some mistakes, such as "more than a million of of years".
- Labels in Figure 1 are probably mixed up - (a) is for the right one, (b) is for the left one?
- I think Figure 11 is not necessary for the paper.
- Table 2 of the Appendix (the core of the paper, but needs a bit of polishing):
- "RFid" should be "RFID".
- Separators (e.g. using double line) could be used between Device Types, e.g. when all the lines of Ultrasound end and Wifi begins, etc.
- Headers could be repeated in each page, so that the reader does not need to scroll back to the first page of Table 2.

Reviewer 2 Report

Overall the paper is a good contribution to the filed of gesture recognition, the text is well written by I'd like to give the following comments:

  • The paper is a nice survey on the recognition of gestures, unfortunately pattern recognition aspects are not clearly addressed. They are mentioned, but  as one can see form the organisation of the paper, it is driven by the sensors not by pattern recognition methodologies. Therefore the papers fits much better the the MDPI-Sensors Journal.
  • The discussion of the paper is too weak, e.g. a discussion  of pattern recognition methods in the context of gestures is missing.  Particularly, gestures are represented by  spatio-temporal or vectorial patterns, this or other ideas could be discuss in terms of information theoretical aspects.

My suggestion is to transfer this paper to another Journal, eg. MDPI-Sensors.